# *Candida parapsilosis* Cell Wall Proteome Characterization and Effectiveness against Hematogenously Disseminated Candidiasis in a Murine Model

**DOI:** 10.3390/vaccines11030674

**Published:** 2023-03-16

**Authors:** Xiaolong Gong, Vartika Srivastava, Previn Naicker, Amber Khan, Aijaz Ahmad

**Affiliations:** 1Clinical Microbiology and Infectious Diseases, School of Pathology, Health Sciences, University of the Witwatersrand, Johannesburg 2193, South Africa; 2NextGen Health, Council for Scientific and Industrial Research, Pretoria 0184, South Africa; 3The Scintillon Institute, 6404 Nancy Ridge Drive, San Diego, CA 92121, USA; 4Infection Control, Charlotte Maxeke Johannesburg Academic Hospital, National Health Laboratory Service, Johannesburg 2193, South Africa

**Keywords:** *Candida parapsilosis*, cell wall proteins, BALB/c mice, vaccine development

## Abstract

*Candida parapsilosis* poses huge treatment challenges in the clinical settings of South Africa, and often causes infections among immunocompromised patients and underweight neonates. Cell wall proteins have been known to play vital roles in fungal pathogenesis, as these are the first points of contact toward environments, the host, and the immune system. This study characterized the cell wall immunodominant proteins of pathogenic yeast *C. parapsilosis* and evaluated their protective effects in mice, which could add value in vaccine development against the rising *C. parapsilosis* infections. Among different clinical strains, the most pathogenic and multidrug-resistant *C. parapsilosis* isolate was selected based on their susceptibility towards antifungal drugs, proteinase, and phospholipase secretions. Cell wall antigens were prepared by β-mercaptoethanol/ammonium bicarbonate extraction from selected *C. parapsilosis* strains. Antigenic proteins were identified using LC–MS/MS, where 933 proteins were found, with 34 being immunodominant. The protective effect of the cell wall immunodominant proteins was observed by immunizing BALB/c mice with cell wall protein extracts. After the immunization and booster, the BALC/c mice were challenged with a lethal dose of *C. parapsilosis*. In vivo results demonstrated increased survival rates and lower fungal burden in vital organs in the immunized mice compared to the unimmunized mice, thereby confirming the immunogenic property of cell wall-associated proteins of *C. parapsilosis*. Therefore, these results advocated the potential of these cell wall proteins to act as biomarkers for the development of diagnostic assays and/or vaccines against infections caused by *C. parapsilosis*.

## 1. Introduction

Infections, as the result of fungal pathogens, have been on the increase globally in recent decades, often in the form of nosocomial infection among hospital patients with compromised immune systems often due to AIDS, surgery, or cancer treatment. *Candida* species are well-known opportunistic pathogens that cause nosocomial infection worldwide, and invasive *Candida* infection (candidiasis) is the most prevalent in the world [1]. Candidiasis often results in blood infection (candidemia), which has a high mortality rate of above 40% [1,2]. In recent decades, infections caused by non-candida species such as *Candida glabrata*, *Candida parapsilosis*, *Candida tropicalis*, and *Candida krusei* have become more prevalent [3]. 

*C. parapsilosis* has become the most common non-candida species in South Africa, where there is also an alarming prevalence of azole-resistant *C. parapsilosis* infection [4], which causes outbreaks of neonatal bloodstream infections in ICU [5]. The azole resistance in *C. parapsilosis* is associated with a Y132F substitution within the ERG11 gene [6]. Underweight neonates are known to be more likely infected by *C. parapsilosis*, where the risk factors consist of horizontal transmissions, such as contamination via medical devices or fluids, hands of health care workers, prosthetic devices, and catheters [7]. 

The cell wall of a fungal pathogen is a vital organelle providing cellular strength and protection from the environment, and also has important roles in pathogen–host interaction including attachment to host tissues, biofilm formation, invasion, host immune recognition, immune evasion, and proteolytic activities [8]. The components of the *Candida* cell wall consist mostly of polysaccharides such as mannan, β-glucan, and chitin [9]. Disruption of cell wall components such as loss of proper N-linked mannosylation in *C. parapsilosis* cell wall due to knockout of the OCH1 gene (that encodes for a Golgi-resident a 1,6-mannosyltransferase) has been shown to cause slower growth, a clumpy cell phenotype, abnormal morphogenesis, defects in cell wall composition, irregular porosity, lower fitness, reduced virulence, and stronger immune response from human peripheral blood mononuclear cells though it does not affect phagocytosis by macrophages [10]. 

Due to all above mentioned information, treatment of *C. parapsilosis* infection is difficult. However, a vaccine can be a suitable alternative solution and experts have been calling for vaccines against fungal pathogens [11,12]. Currently, there is still no licensed vaccine against fungal pathogens available [13]. The goal of the current pilot study is to characterize immunodominant cell wall proteins of *C. parapsilosis* and observe its protective effect, which will be helpful for developing potential diagnostic biomarkers and vaccine candidates.

## 2. Materials and Methods

### 2.1. Candida Species

Twelve *C. parapsilosis* strains were isolated from clinical settings and stored under −80 °C in the Department of Clinical Microbiology and Infectious Diseases, Faculty of Health Sciences, University of the Witwatersrand. The samples were stored inside 100% glycerol stock. We revived them by dropping 10 µL of glycerol stock onto Sabouraud dextrose agar (SDA) plates and incubating the plates for 48 h under 37 °C. All the strains were handled following the laboratory SOPs to avoid any contamination and cross-contamination. For biosafety, the protocol was assessed and approved by the Institutional Biosafety Committee under the reference number 20210502Lab. 

### 2.2. Antifungal Susceptibility Profiling

We calculated the minimum inhibitory concentration (MIC) and the minimum fungicidal concentration (MFC) of the twelve clinical strains of *C. parapsilosis* using Fluconazole (FLZ; Sigma-Aldrich, St. Louis, MO, USA), Amphotericin B (AmB; Sigma-Aldrich, USA), Nystatin (NYST; Sigma-Aldrich, USA), and Caspofungin (CAS; Sigma-Aldrich, USA). A broth microdilution assay was performed under the guidelines recommended in the standard M27 document (4th ed.) presented by the Clinical and Laboratory Standard Institute [14]. The stock solution of the prior mentioned drugs was prepared using dimethyl sulfoxide (DMSO; Merck, RSA) and the test concentration ranged from 1250 to 9.77 µg/mL for FLZ, 62.5 to 0.49 µg/mL for AmB, 250 to 19.5 µg/mL for NYST, and 8 to 0.06 µg/mL for CAS. Sterility and growth control were included for comparison. The plates were incubated at 37 °C for 48 h. The MIC results were determined by visual observation, where antifungal drugs with the lowest concentration inhibited the fungal growth when compared to the growth control was considered as MIC value.

For estimating MFC, 10 µL from the wells with different concentrations of antifungal drugs were subcultured onto Sabouraud dextrose agar (SDA; Merck, RSA) plates followed by incubation at 37 °C for 48 h. The MFC was determined by observing the lowest concentration with no growth on the agar plate [15].

### 2.3. Extraction of Cell Wall Proteins

Cell wall proteins were extracted as described previously by El Khoury and co-workers with minor modifications [16]. *C. parapsilosis* MRU# 4112 was selected for cell wall protein extraction based on the antifungal susceptibility results. The concentration of *C. parapsilosis* MRU# 4112 cells was adjusted to 0.5 McFarland in 40 mL and was then inoculated in 2 L liquid mediums (2 to 100 mL ratio). The cells were incubated for 48 h in Sabouraud dextrose broth (SDB). Afterward, the cells were centrifuged at 4500 rpm for 3 min and were washed with PBS twice, resulting in an average of 7 g/L of *C. parapsilosis* cell pellet. Later on, the pellet was resuspended in 5 mL Tris (5 mM pH 7.8). A protease inhibitor cocktail (6 µL; Sigma-Aldrich, USA) was added along with glass beads (0.45–0.55 µm, Sigma-Aldrich, USA) at a 1:1 ratio to the pellet and vortexed for 10 cycles with 1 min on vortex followed by 1 min on ice. Thereafter, beads were removed by transferring the solution into another pre-weighed falcon tube. The remaining beads were washed several times with cold NaCl (1 mM) until the supernatant becomes transparent and mixed again in the respective falcon tubes without transferring any beads during the procedure. The samples were spun at 3000 rpm for 5 min and the supernatant was discarded and the pellet was resuspended in 40 mL 1M NaCl solution and spun at 3000 rpm for 5 min, repeated 3 to 4 times. Later on, protein extraction buffer (50 mM Tris; 2% SDS; 100 mM Na-EDTA; 150 mM NaCl; pH, 7.8) with beta-mercaptoethanol (β-ME), 8 µL per 1 mL PEB, was used to resuspend the washed pellet, 0.5 mL per 100 mg net weight cellular pellet followed by boiling for 10 min. The mixture was spun for 5 min at 3000 rpm; thereafter, the supernatants containing SDS extractables were collected. This procedure was performed twice. Acetone, at a 1:1 ratio to the supernatant, was added to the sample and incubated overnight under −20 °C. The next day, the samples were centrifuged at 4400 rpm for 5 min and the supernatant was poured out. The pellets were washed three times with acetone and left to dry. To determine the protein concentration, the dried protein powder was resuspended in protein lysis buffer [urea, 7 M; thiourea, 2 M; CHAPS, 4 % (*w*/*v*); pharmalyte, 0.8 % (*v*/*v*) (Thermo Fisher Scientific, Waltham, MA, USA)] and protein concentration was estimated using a DC Protein Assay (Bio-Rad), with BSA as a standard.

### 2.4. Sample Preparation for LC–MS Analysis

Cell wall extract samples were incubated with Bolt™ LDS sample buffer and sample reducing agent (Invitrogen, Waltham, MA, USA) and electrophoresed on a Bolt™ 4 to 12%, Bis-Tris, 1.0 mm, Mini Protein Gel (Invitrogen, USA). The polyacrylamide gels were stained with GelCode™ Blue Stain Reagent (Thermo Fisher Scientific, USA) and de-stained with Milli-Q water. Proteins were digested from gel fractions according to 1. Each sample was prepared in 6 separate gel fractions (based on approximated molecular weight ranges: >100, 60–100, 40–60, 25–40, 15–25, and <15 kDa). Briefly, the proteins were reduced in gel with 10 mM DTT in 25 mM AMBIC for 1 h at sixty degrees. Samples were cooled to room temperature, then 100% acetonitrile was added and incubated for 10 min. The supernatant was discarded and 55 mM iodoacetamide (IAA) in 25 mM AMBIC was added to the gel pieces. The reaction proceeded in the dark for 20 min at room temperature. The supernatant was discarded, and gels were dehydrated with 25 mM AMBIC in 50% acetonitrile, vortexed and the supernatant was removed. The gel pieces were dried to completeness and freshly prepared trypsin was added; protein digestion was allowed to proceed overnight at 37 °C. The digestion was quenched by adding formic acid to a final of 0.1% and the samples were dried under a vacuum. Dried samples were re-suspended in 2% acetonitrile and 0.2% formic acid for mass spectrometry analysis.

### 2.5. LC–MS Data Acquisition

Tryptic peptides from each gel fraction were analyzed using a Dionex Ultimate 3000 RSLC system coupled to an AB Sciex 6600 TripleTOF mass spectrometer. Injected peptides were inline de-salted using an Acclaim PepMap C18 trap column (75 μm × 2 cm; 2 min at 5 μL·min 1 using 2% ACN/0.2% FA). Trapped peptides were gradient eluted and separated on a Waters nanoEase CSH C18 column (75 μm × 25 cm, 1.7 µm particle size) at a flow rate of 0.3 µL·min 1 with a gradient of 10–55% B over 10 min (A: 0.1% FA; B: 80% ACN/0.1% FA). The 6600 TripleTOF mass spectrometer was operated in the positive ion mode. Data-dependent acquisition (DDA) was employed; precursor (MS) scans were acquired from *m*/*z* 400 to 1500 (2+–5+ charge states) using an accumulation time of 100 ms followed by 40 fragment ion (MS/MS) scans, acquired from *m*/*z* 100 to 1800 with 20 ms accumulation time each.

### 2.6. LC–MS Database Searching

Raw data files (.wiff) were searched with Protein Pilot V5.0 software (SCIEX), using a database containing sequences from *C. parapsilosis* reference proteome (downloaded from UniProt on 24 June 2021) and common contaminants. Trypsin was set as the digestion enzyme, cysteine alkylation (iodoacetamide) was allowed as a fixed modification and biological modifications were allowed in the search parameters. A 1% false discovery rate filter was applied at the protein level for the refinement of identifications.

### 2.7. Bioinformatic and Functional Analyses

The list of gene names for the identified proteins was used to obtain the protein sequences (ORF translation) from the *Candida* Genome Database (http://www.candidagenome.org/, accessed 13 December 2022) 2 using the CGD Batch Download tool and gene ontology enrichment analysis performed using the CGD GO Slim Mapper tool using the default settings. The complete list of sequences was submitted to the VaxiJen v2.0 server (http://www.ddg-pharmfac.net/vaxijen/VaxiJen/VaxiJen.html) 3 to predict the presence of protective antigens, a threshold score of 0.9 was applied. The DeepLoc-1.0 (https://services.healthtech.dtu.dk/service.php?DeepLoc-1.0) 4 server was used to accurately (profile setting) predict subcellular localization of the predicted antigens.

### 2.8. In Vivo Study

BALB/c mice, 8 to 10 weeks old, were used for this study. The mice were acclimatized for a week and weighted weekly (18–21 g). All animal studies were performed according to the protocol for the care and use of animals sanctioned by the Animal Research Ethics Committee (AREC), University of the Witwatersrand, Johannesburg, South Africa (AREC- 2022/01/01/C). The protocol was inspected and approved by the Institutional Biosafety Committee (IBC) and the Department of Agriculture, Land Reform and Rural Development (DALRRD).

### 2.9. Immunization

A total of 36 BALB/c mice were used in the animal model. The mice were randomly separated into 3 groups: (a) the healthy control group (*n* = 12), (b) the infection control group (*n* = 12), and (c) the immunization group (*n* = 12). The control group was mock immunized with 0.2 mL of sterile phosphate buffer saline whereas, in order to ensure the injection of a sufficient quantity of protein extract that may induce an appropriate immune response against *C. parapsilosis* infection, the test groups were injected with 300 µg of cell wall protein extract via the intraperitoneal route [17]. However, the protein extract was not supported with any adjuvant system. Thereafter, a booster dose of the same protein concentration was given on day 21. Afterward, all the mice, except the healthy control group, were infected with a lethal injection of *C. parapsilosis* MRU# 4112 (5 × 10^6^ CFU/mL) (as estimated by using MicroScan Turbidity meter, Beckman Coulter, in a total volume of 150 μL of sterile saline) via the intravenous route. Plate counts were performed for confirmation of the cell count and viability of infecting inoculum in each mouse. The protective effect was observed for 3 weeks after the infection by monitoring the survival rate of mice in the vaccinated group.

### 2.10. Histopathology

After observing the protective effect for 3 weeks, all the mice were sacrificed according to the SOP followed by the Wits Research Animal Facility. After the euthanization of mice, kidneys, lungs, livers, and spleens were removed aseptically and weighed. The collected organs were then fixed and stored in 10% buffered formalin. The fixed tissues were embedded in paraffin followed by cutting a longitudinal section of 4 µm and stained with Periodic acid-Schiff (PAS). The stained sections were then observed under a microscope (Leica Microsystems, Heerbrugg, Switzerland).

The images of the samples were taken under a lens of 100× magnification. Day light blue filter was used to balance out the color of the light source. LED was used as the light source. Color corrections were performed for images that were too yellow or blue.

The collected organs were homogenized in sterile PBS (chilled) and the processed homogenate was then seeded onto SDA plates containing 100 µg/mL ampicillin. The plates were incubated at 37 °C for 48 h and afterward, colonies were counted, and the result was recorded as log_10_ CFU/mg of tissue for estimating the fungal burden.

### 2.11. Statistical Analysis

The results of fungal burden were analyzed using a log-rank test and the unpaired Student’s *t*-test (two tailed) in GraphPad Prism software, version 8.0.1. Statically significant differences were defined as *p*-value.

## 3. Results

### 3.1. Antifungal Susceptibility Profiling

Antifungal susceptibility profiling of the twelve *C. parapsilosis* strains was performed against four commonly used antifungal drugs under the CLSI-recommended guidelines. The results obtained from the broth microdilution assay, MIC, and MFC values for FLZ, AmB, NYST, and CAS were recorded in Table 1. According to the MIC and MFC results, *C. parapsilosis* MRU# 4112 was selected for further analysis.

### 3.2. Proteomic Analysis of the Cell Wall-Associated Proteins and Protective Antigen Prediction

Liquid chromatography–mass spectrometry (LC–MS) identified 933 protein groups for cell wall extract of *C. parapsilosis* MRU# 4112 strain. Gene set-enrichment analysis showed a high frequency of proteins involved in the regulation of biological processes (194 proteins, 20.8%), transport (182 proteins, 19.5%), and organelle organization (168 proteins, 18%). The enrichment distribution of all identified proteins in the cell wall extract is shown in Figure 1. Other significant biological processes that were observed at a low frequency are cell adhesion (3 proteins: protein transport protein SEC31, agglutinin-like protein 7, and non-specific serine/threonine protein kinase), biofilm formation (3 proteins: non-specific serine/threonine protein kinase, fatty acid synthase subunit alpha, and HTH APSES-type domain-containing protein) and response to drugs (2 proteins: E3 ubiquitin-protein ligase and CRAL-TRIO domain-containing protein).

### 3.3. Functional Analysis of Predicted Antigens of C. parapsilosis MRU# 4112 Strain Cell Wall Extract

Thirty-four of the identified proteins were predicted to be protective antigens (Table 2). The predicted antigens were further annotated by determining the subcellular localization using the DeepLoc 4 prediction tool. Five antigenic proteins were predicted to occur in the membrane form or localized to the extracellular space: *Candida* ALS N domain-containing protein and four uncharacterized proteins. Wewe collated the *C. albicans* orthologs of each of these antigenic proteins to provide further functional annotation (Table 3). However, three of the ortholog proteins are also uncharacterized in *C. albicans* strains.

### 3.4. In Vivo Studies

#### Effectiveness of Immunization

The lethal dose of *C. parapsilosis* challenge (5 × 10^6^ CFU/mL) did not induce 100% mortality in the infection control group. However, mice in the infection control group displayed signs of distress such as orbital tightening, rough hair coat, and gradual weight loss after 48 and 72 h of infection, which was used as a sign of morbidity that depends on both fungal virulence and the host’s inflammatory immune response. Several studies have used this marker for disease severity in mice with invasive candidiasis or other types of infection [18,19,20,21]. According to the protocol, we had to sacrifice the mice when such signs of distress appear. These results are similar to another study for analyzing the protective effect of *C. parapsilosis* cell wall mutant [10]. Mice (*n* = 4) from the healthy control, infection control, and immunization group were sacrificed after 48 and 72 h of infection to further investigate the protective effects of cell wall proteins, while the remaining mice were observed until the end of the experiment (21 days) (Figure 2). According to plate count, the CFU on the plates for the immunization group was significantly lower than the infection control group (Figure 3).

### 3.5. Histopathology

A higher *C. parapsilosis* cell distribution can be observed in the liver, lungs, and spleen tissue sections of the infection control mouse, while damaged tissues can be observed in the kidneys. Whereas significantly less fungal burden was observed in the mouse immunized with cell wall protein. The *C. parapsilosis* was presented only in the yeast form and this finding was in agreement with previous studies [21,22] Therefore, the results advocated the protective effect of the cell wall proteins against *C. parapsilosis* infection (Figure 4).

## 4. Discussion

*C. parapsilosis* has developed a trend of increasing antifungal drug resistance in recent years. Therefore, it has become paramount to develop a vaccine for protection against infection. However, there is currently no available vaccine against fungal infection. In this study, we extracted *C. parapsilosis* MRU# 4112 cell wall protein and injected it into BALB/c mice to prevent *C. parapsilosis* infection.

According to the functional analysis, 933 proteins were found, and 34 out of 933 proteins were predicted to be antigenic. Five of the identified antigenic proteins were identified as membrane or extracellular proteins (bold font). However, four out of the five were uncharacterized proteins (Table 1). The only characterized protein is located on the cell membrane, CPAR2_404800, which encodes *Candida* ALS N domain-containing protein.

The CPAR2_404800 gene is homologic to ALS7/C3_06320W_A in *C. albicans* SC5314, which encodes the *C. albicans* ALS7 protein. The agglutinin-like sequence (ALS) family protein consists of eight discovered genes that encode large glycosylphosphatidylinositol (GPI)-linked glycoproteins on the cell surface and is well known to play an important role in adhesion to the host cells in *C. albicans* [23] and *Saccharomyces cerevisiae* [24], which is vital for pathogenesis as adhesion toward human epithelial cells can significantly increase the risk of invasive Candidiasis. The ability of adherence to biotic and/or abiotic surfaces is also an important function of the cell wall proteins, which allows the fungus to colonize the host and/or medical devices. The disruption of this gene in *C. parapsilosis* can result in a reduction in adhesion and virulence [25]. The *C. parapsilosis* ALS7 protein has a strong capacity for binding to plasminogen and high-molecular-mass kininogen, which are essential components of the human fibrinolytic system and proinflammatory/procoagulant contact-activated kinin-forming system, respectively [26]. The disruption of these two systems can result in the development of systemic inflammatory response syndrome (SIRS), septic shock, and severe sepsis often associated with critical conditions and death [27,28].

The other four proteins, namely CPAR2_405130, CPAR2_701270, CPAR2_703940, and CPAR2_805040, were uncharacterized in *C. parapsilosis*. All these uncharacterized proteins were discovered a long time ago, but not much follow-up research performed in the *Candida* species, especially *C. parapsilosis*. Now that we have predicted that these proteins can play important role in raising host immune response against *C. parapsilosis*, more research should be conducted to further investigate the immunomodulatory aspect of these proteins.

CPAR2_405130 is homologic to *C. albicans* C3_02270W; however, its function is uncharacterized among the *Candida* species. Its gene ortholog is SPAC26A3.14c in *Schizosaccharomyces pombe*, which is a hypothetical protein with a low UV sensitivity [29,30].

CPAR2_701270 is orthologous to *C. albicans* PHO88/CR_09320C and *C. glabrata* PHO88/CAGL0K12276g. PHO88/CR_09320C is a *C. albicans* membrane proteome that plays roles in phosphate transportation and biofilm-regulated expression, while Amphotericin B can repress its function [31,32,33]. Although PHO88/CAGL0K12276g was discovered in *C. glabrata* with high similarity to *Saccharomyces cerevisiae*, its function is assumed to be involved in the PHO signal pathway during phosphate starvation, similar to the one in *S. cerevisiae* [34].

CPAR2_703940 is orthologous *Saccharomyces cerevisiae* PRB1 to *C. albicans* C7_03860W and *C. glabrata* CAGL0B03619g. In *S. cerevisiae* PRB1 is the gene that encodes the precursor of a vacuolar protease that functions as a proteinase that degrades and overturns proteins during limiting nutrient and stressful conditions [35,36,37,38]. The gene is repressed by glucose, indicating PRB1 expression is activated when an alternative source of carbon is needed for synthesizing ATP [39]. The PRB1 gene is also required for Histone H3 N terminal truncation in yeast cells [40]. The ortholog C7_03860W was characterized in *C. albicans* as a vacuolar protease [41]. The gene is expressed when *C. albicans* encounter polymorphonuclear leukocytes such as neutrophils, possibly due to downregulating of *C. albicans* protein synthesis and reduced proliferation as a result of a lack of nutrients [42]. The ortholog CAGL0B03619g is expressed in *C. glabrata* during osmotic stress [43]. In both cases, the orthologs have a similar function to the PRB1 in *S. cerevisiae*.

CPAR2_805040 is characterized as a secretory protein located in *C. parapsilosis* cell wall with an unknown function [44]. The alias of the gene, CPAG_03284, has an increased expression in azole-resistant *C. parapsilosis* strains [45]. The gene is orthologous to *C. albicans* C1_10170W, which was discovered during an antifungal compound experiment [46]. However, the function of this *C. albicans* extracellular secretome is currently unknown [47].

Furthermore, proteins such as CPAR2_405130, 701270, 703940, and 805040 found in the protein extracts were characterized as moonlight proteins. Notably, moonlight proteins have more than one function in *Candida* species; although they originated in the cytoplasm, they can be found in cell walls with a different purpose and plays a vital role in the pathogenicity of *Candida* species. Studies have shown these moonlight proteins have the potential to be used as a vaccine against *C. albicans* and *C. glabrata* infection [48].

The orthologous of the protein encoded by CPAR2_404800, CPAR2_701270, CPAR2_703940, and CPAR2_805040 are mostly reported for one or the other pathogenic attributes in different species *Candida* and *S. cerevisiae* and therefore, the presence of these virulence-associated proteins in cell wall extract of *C. parapsilosis* further advocate its immunogenic property. Moreover, we speculate that these proteins are immunogenic in nature and together they might have the potency to trigger a strong immune response against the *C. parapsilosis* in the vaccinated group. Most importantly, the antigens from the cell wall proteins of *Candida* species have the ability to activate the adaptive immune response including Th1 (type 1 helper T-cell) and Th17 cell immune responses, antigen presenting cells (APCs), B cells and T cells during infection to reduce fungal burden [48,49,50,51,52,53]. The APCs (such as macrophages and dendritic cells) when come in contact with the cell wall proteins release cytokines IFN-γ and IL-2 to activate Th1 response, which in turn leads to macrophages activation to release IFN-γ and IL-12 to activate more macrophages and Th1 [51]. Additionally, the cell wall proteins also stimulate Th17 differentiation which releases IL-17 to prolong the survival of neutrophils and activate and recruit neutrophils to the infection site to engulf *Candida* cells and induce inflammation [51]. The APCs also present cell wall antigens to the CD4+ T cells and CD8+ T cells, the CD4+ T cells then present to the B cells to develop into plasma cells and create antibodies [52,53]. Alternatively, it is also possible that the immune response mostly involved the components of the innate immune system. Where myeloid cells including peripheral blood mononuclear cells, peripheral blood mononuclear cell-derived macrophages, polymorphonuclear neutrophils, DCs, and bone marrow-derived macrophages all involved in the innate immune response against *C. parapsilosis* infection; which are triggered by the non-protein components of *C. parapsilosis* cell walls [54]. Correspondingly, the *C. parapsilosis* infection activates the peripheral blood mononuclear cells to release pro-inflammatory cytokines including IL-1β, IL-6, and tumor necrosis factor alpha, as well as anti-inflammatory cytokines IL-10 and eliminate fungal yeast cells via phagocytosis. Moreover, due to a lack of research and published data in *C. parapsilosis*, we conjectured that these proteins together might play important role in activating host immune response in a similar manner as observed in other *Candida* species. The cell wall protein extract appeared to protect BALB/c mice from *C. parapsilosis* infection which was further supported by the histopathology results (Figure 4). In another study, Pérez-García et al. (2016) [10] using an animal model found that BALB/c mice infected by *C. parapsilosis* och1Δ null mutant had a significant reduction in fungal burden in the spleen, kidneys, and liver at 1, 2, and 7 days post infection comparing to WT *C. parapsilosis*, which is very similar to our results of using cell wall proteins to protect mice against infection. Furthermore, in the kidneys, we did not see individual *C. parapsilosis* cells as clearly as in the other organs; however, we do see the damages caused by the infection were mild in the immunized mice as compared to the infection control group. In all other organs (the spleen, liver, and lung), scattered individual cells can be observed in the immunized mouse. Where significantly more cells and even clusters of *C. parapsilosis* cells can be observed in the infection control mouse. We can see that the *C. parapsilosis* cells infected the periphery of the lungs). We have also used VITEK 2 System version 9.02 to identify the colonies that grew on the plates for fungal burden analysis. The result indicates that the colonies were *C. parapsilosis* (Appendix A).

## 5. Conclusions

The pilot study has successfully shown that the antigens from extracted cell wall proteins of *C. parapsilosis* MRU# 4112 can be identified using mass spectrometry and the immunodominant proteins can protect mice from infections. A total of 34 immunodominant proteins were identified, including CPAR2_405130, CPAR2_701270, CPAR2_703940, and CPAR2_805040, which are new immunodominant proteins that have not been characterized in *C. parapsilosis* yet. A few were characterized in *C. albicans* that function in stress response, resistance, and pathogenicity pathways (CPAR2_404800, CPAR2_701270, and CPAR2_703940). It is hypothesized that the cell wall proteins induced antibodies in mice, which prevented disseminated *Candida* infection. The results have showed that the characterized immunodominant proteins have the potential to be used as diagnostic biomarkers and vaccines against *C. parapsilosis* infection. However, more studies need to be conducted such as purification of these immunodominant proteins and applied to in vivo models to identify and evaluate the protective potential of individual proteins against different *Candida* infections.

## Figures and Tables

**Figure 1 vaccines-11-00674-f001:**
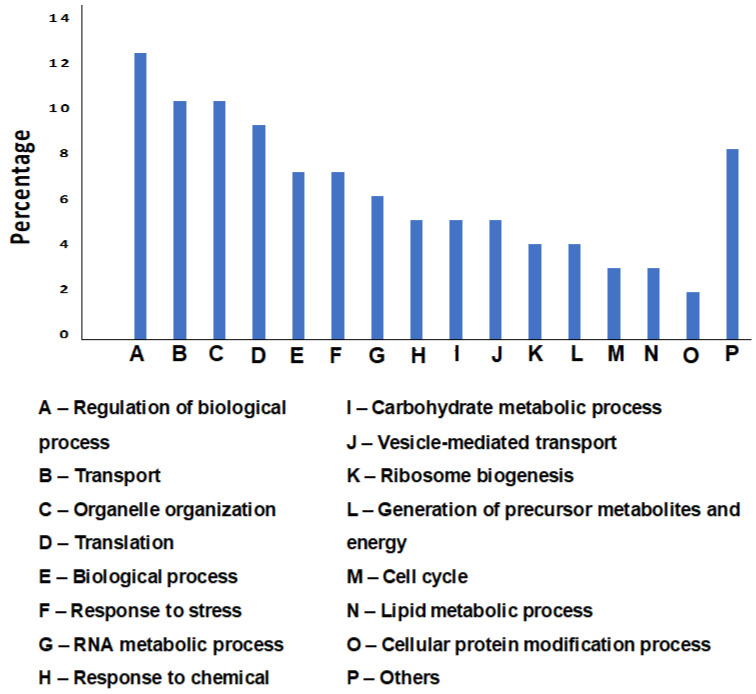
Bar graph showing gene ontology enrichment analysis of biological processes for the proteins identified in the cell wall extract. Biological processes that were common to <5% of the proteins were excluded from the chart.

**Figure 2 vaccines-11-00674-f002:**
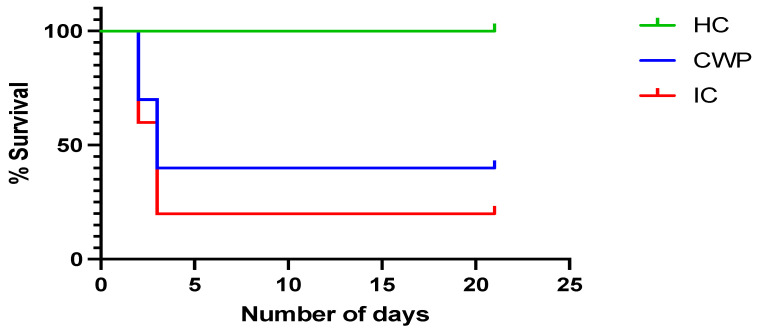
Survival probability of cell wall protein immunized BALB/c mice. The survival proportion states the potency of cell wall protein extract in protection against infection caused *C. parapsilosis*. Mantel–Cox log-range test was used to compare the survival in different groups. HC = health control; CWP = cell wall protein; IC = infection control. *p* value < 0.005.

**Figure 3 vaccines-11-00674-f003:**
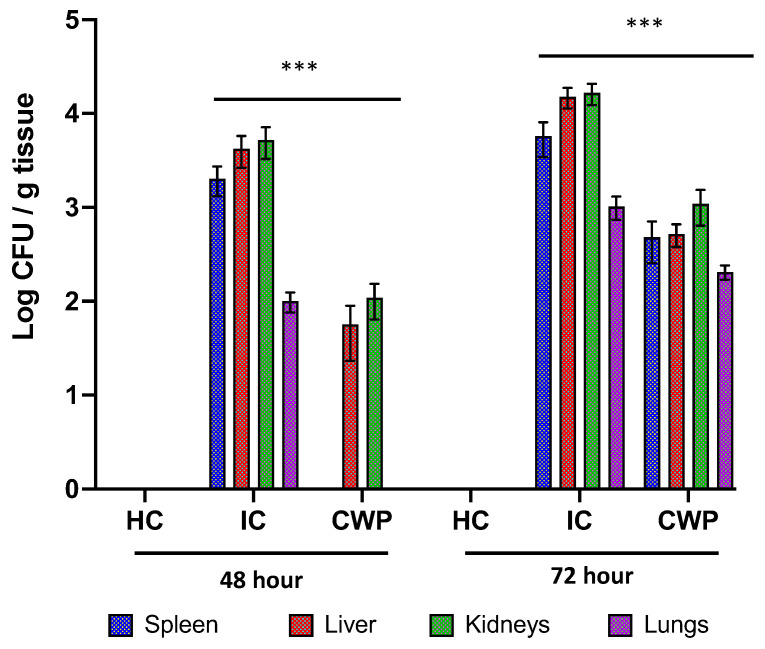
Analysis of fungal burden. The effect of cell wall protein extract against infection caused by *C. parapsilosis*. The fungal burden was expressed as geometric means of logarithmic values for CFU/mg of tissues. HC = health control; CWP = cell wall protein; IC = infection control. (*** *p* < 0.05).

**Figure 4 vaccines-11-00674-f004:**
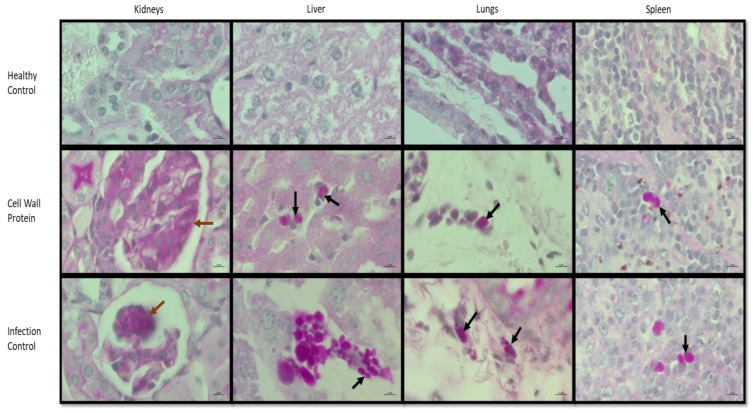
Comparative analysis of histopathology of different tissue sections from healthy control, immunized, and *C. parapsilosis*-infected mice after 48 h. Tissue sections recovered post infections were stained with PAS. The mouse from the infection control group displayed a higher fungal burden. While the cell wall protein immunized mouse showed significantly reduced fungal burden. The black arrows indicate of *C. parapsilosis* cells in the target tissues while the brown arrows indicate the damages in the kidney.

**Table 1 vaccines-11-00674-t001:** Antifungal susceptibility profiling.

Strain	01IS MRU#: 4315		02IS MRU#: 4282		03IS MRU#: 4090		04IS MRU#: 4112	
µg/mL	MIC	MFC	MIC	MFC	MIC	MFC	MIC	MFC
FLZ	312.500	FS	625.000	FS	312.500	FS	312.500	FS
AmB	0.977	1.953	0.977	1.953	0.977	1.953	31.250	62.500
NYST	62.500	125.000	31.250	FS	62.500	FS	125.000	FS
CAS	0.125	0.250	0.125	0.250	0.125	0.250	0.500	1.000
Strain	05IS MRU#: 4120		06IS MRU#: 4334		07IS MRU#: 4033		08IS MRU#: 4108	
µg/mL	MIC	MFC	MIC	MFC	MIC	MFC	MIC	MFC
FLZ	312.500	FS	312.500	FS	156.250	FS	312.500	FS
AmB	1.953	31.250	1.953	7.813	3.906	62.500	15.625	31.250
NYST	62.500	250.000	125.000	FS	62.500	FS	31.250	FS
CAS	0.250	0.500	0.125	0.500	0.250	2.000	0.125	0.250
Strain	09IS MRU#: 4036		10IS MRU#: 4086		11IS MRU#: 5473		12IS MRU#: 5503	
µg/mL	MIC	MFC	MIC	MFC	MIC	MFC	MIC	MFC
FLZ	312.500	FS	312.500	FS	625.000	FS	312.500	FS
AmB	7.813	31.250	7.813	31.250	7.813	62.500	1.953	31.250
NYST	31.250	62.500	15.625	31.250	7.813	125.000	31.250	125.000
CAS	0.500	2.000	0.063	0.125	0.125	0.250	1.000	4.000

The concentrations are in µg/mL. FS = Fungistatic.

**Table 2 vaccines-11-00674-t002:** List of predicted antigenic proteins that are present in the cell wall extract of *C. parapsilosis*.

Gene	Protein Names	Localization	Type
CPAR2_100640	Uncharacterized protein	Cytoplasm	Soluble
CPAR2_104370	Uncharacterized protein	Nucleus	Soluble
CPAR2_104690	GTP-binding nuclear protein	Cytoplasm	Soluble
CPAR2_105980	Uncharacterized protein	Cytoplasm	Soluble
CPAR2_106320	Uncharacterized protein	Cytoplasm	Soluble
CPAR2_109010	Uncharacterized protein	Plastid	Soluble
CPAR2_110110	HTH cro/C1-type domain-containing protein	Nucleus	Soluble
CPAR2_201550	Glycine cleavage system H protein	Mitochondrion	Soluble
CPAR2_203410	Uncharacterized protein	Mitochondrion	Soluble
CPAR2_204240	Proteosome_alpha_1 domain-containing protein	Nucleus	Soluble
CPAR2_205620	Ribosomal_S10 domain-containing protein	Cytoplasm	Soluble
CPAR2_207050	Uncharacterized protein	Cytoplasm	Soluble
CPAR2_211070	NET domain-containing protein	Nucleus	Soluble
CPAR2_212330	Uncharacterized protein	Cytoplasm	Soluble
CPAR2_300430	Uncharacterized protein	Nucleus	Soluble
CPAR2_302060	Histone-glutamine methyltransferase	Nucleus	Soluble
CPAR2_304080	Adenylyl-sulfate kinase	Cytoplasm	Soluble
CPAR2_401210	Proliferating cell nuclear antigen	Nucleus	Soluble
CPAR2_403290	Uncharacterized protein	Cytoplasm	Soluble
**CPAR2_404800**	**Candida ALS N domain-containing protein**	**Cell surface**	**Membrane**
**CPAR2_405130**	**Uncharacterized protein**	**Mitochondrion**	**Membrane**
CPAR2_503400	Pre-mRNA-processing protein 46	Nucleus	Soluble
CPAR2_503520	Branched-chain-amino-acid aminotransferase	Cytoplasm	Soluble
CPAR2_504170	40S ribosomal protein S24	Cytoplasm	Soluble
CPAR2_600900	Uncharacterized protein	Cytoplasm	Soluble
CPAR2_601260	CS domain-containing protein	Nucleus	Soluble
CPAR2_602580	Uncharacterized protein	Cytoplasm	Soluble
**CPAR2_701270**	**Uncharacterized protein**	**Endoplasmic reticulum**	**Membrane**
CPAR2_701500	Uncharacterized protein	Cytoplasm	Soluble
CPAR2_701770	H/ACA ribonucleoprotein complex subunit NOP10	Nucleus	Soluble
**CPAR2_703940**	**Uncharacterized protein**	**Extracellular**	**Soluble**
CPAR2_800880	Actin-related protein 2/3 complex subunit 4	Cytoplasm	Soluble
CPAR2_802120	ANAPC4_WD40 domain-containing protein	Nucleus	Soluble
**CPAR2_805040**	**Uncharacterized protein**	**Extracellular**	**Soluble**

All membrane and extracellular proteins are shown in bold.

**Table 3 vaccines-11-00674-t003:** Details of *C. albicans* orthologs of the *C. parapsilosis* antigenic proteins occurring in the membrane form or localized to extracellular space.

Gene	Protein Names	*C. albicans* Ortholog Gene	*C. albicans* Ortholog Protein Name
CPAR2_404800	*Candida* ALS N domain-containing protein	ALS7/C3_06320W	Agglutinin-like protein 7
CPAR2_405130	Uncharacterized protein	C3_02270W	Uncharacterized protein
CPAR2_701270	Uncharacterized protein	PHO88/CR_09320C	Inorganic phosphate transport protein PHO88
CPAR2_703940	Uncharacterized protein	C7_03860W	Uncharacterized protein
CPAR2_805040	Uncharacterized protein	C1_10170W	Uncharacterized protein

## Data Availability

The mass spectrometry proteomics data have been deposited in the ProteomeXchange Consortium via the PRIDE partner repository.

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
