# Peer review of "Candida parapsilosis Cell Wall Proteome Characterization and Effectiveness against Hematogenously Disseminated Candidiasis in a Murine Model"

_vaccines, 2023, doi:10.3390/vaccines11030674_

Round 1

Reviewer 1 Report (Previous Reviewer 3)

In Figure 4, authors should move the pointer(s) closer to the targeted object, in  particular for the liver and kidney (of both infection control group and cell wall group), where "clusters of  darker objects": are more often seen. 

Just point at what you think clearly is the most typical of Candida yeast. Do not try to use one pointer for the whole cluster of darker objects (otherwise you cannot clearly distinguish which one is yeast from which not?).

Figure 4 is no longer a typical PAS stained slide that you see under a light microscope for a regular histopathological examination.  Revised whatever manipulation (added lens, filters, light source etc) you used in section 2.10 after line 208.

Authors should have the confidence to name the pathologist who they consulted for reference.

Author Response

Response: Thank you for your ample input and critical review. We have moved the pointer closer to the damaged areas and yeast cells observed in the images. Unfortunately, in kidney Candida cells are underrepresented as the damage in the cells is more prominent, which is now pointed with brown arrows in the revised figure. We have also added the procedures for image in section 2.10. In the study, we used 100X magnification lens, daylight blue filter, LED light source and performed colour correction for a clearer presentation. We agree that the quality of the Fig 4 is not the best; however, we want to make sure that no unethical manipulation to the image was done. Furthermore, the confidence can be built from the CFU where we have seen a good Candida population growing. We also performed MALDI of the Candida cells obtained from these organs to confirm that the Candida species is same as the animals were infected with. Based on all these results, we have a confidence on what we are claiming in this manuscript. Related to the pathologist, we consulted our Head of Department, Prof. Vindana Chibabhai, Prof. Shabnum Meer (Oral pathologist) and Mrs Hasiena Ali (Principal Technician School of Anatomical Sciences) to confirm that the figure represents yeast cells.

Reviewer 2 Report (Previous Reviewer 1)

Thank you for including the possibility that the protection may have been mediated by the innate response rather than acquired immunity to the proteins inoculated. That need not preclude anti-candida vaccine development however it suggests a therapeutic vaccine rather than a prophylactic one. Testing your vaccine such a use should be on your agenda sooner rather than later.

Author Response

Authors thank reviewer for the suggestions. As mentioned previously, the work is ongoing, and we are only waiting for further funding to continue the other part of this project.

This manuscript is a resubmission of an earlier submission. The following is a list of the peer review reports and author responses from that submission.

Round 1

Reviewer 1 Report

              This article may report an important first step in the making of a needed new vaccine, however, there are significant gaps in the information provided in Materials and Methods and in the analyses performed that make it unacceptable for publication in its current form.

- The Materials and Methods section is supposed to give other scientists enough information about the procedures to allow them to repeat the studies in another laboratory. This is done by either giving clear and complete information about (usually novel or uncommon) procedures or references to other publications in which the procedures are clearly described.

- In this article the only reference in the entire M&M section is to textbook guidelines which were followed “with appropriate adjustment” undescribed.

- A complicated cell wall protein extraction is described (M&M section 2.3) but while volumes of suspension are given no estimate of the number of cells extracted is given so it is impossible to determine the yield of this procedure.

- Furthermore, some of the reagents are listed as “protease inhibitor cocktail” or “protein extraction buffer” with no information about their composition. At one point it says that “The (glass) beads were then removed” but it doesn’t say how; that is important information because if they were filtered out then one wants to know the pore size of the filter to have an idea of what else might have been removed or if they were centrifuged out then at what speed so that the reader has some idea of what else might have been removed.

- In the Immunization section (M&M section 2.9) no information is given about the amount of protein in the inoculum which is simply described as “60 ul of extracted cell wall proteins” although later the article states that “a booster dose of the same protein concentration was given” so the authors must know what the protein concentration was. This is important information because most of the 933 C. parapsilosis proteins detected in the cell wall extract by the sophisticated mass spec analysis described were unlikely to have been present in the inoculum at sufficient quantity to induce an immune response.

- Also, it is not stated whether the extract was inoculated with an adjuvant to enhance the immune response to low concentration or poorly immunogenic proteins. If an adjuvant was used, then it is important to name it and detail the amount used. But I say “if an adjuvant was used” because that raises another and perhaps more important issue with this paper.

- It is well known that pathogen cell walls are replete with molecules that stimulate innate immune responses with adjuvant-like effects (pathogen-associated molecular patterns or PAMPs) and some adjuvants (e.g. MPLA) are actually purified from pathogen extracts, so these authors may have relied on adjuvant-like molecules in their cell wall extract that co-purified with the cell wall protein. In fact, absent information on the proteins present in the extract in sufficient quantity to be immunogenic, it is possible that the only immune response generated was the innate response and there was no appreciable or functional anti-cell-wall-protein antibody response induced. Innate immune mechanisms contribute to the control of fungal pathogens such as C. parapsilosis and the challenge experiment described in this article was performed only two weeks after the booster shot so any innate immunity induced by the inoculations was still active during the challenge and could have been the sole reason for the protection observed. This could be sorted out by repeating the immunization/challenge study with an adjuvant alone control arm (if an adjuvant was used) or a control arm with the cell wall extract treated with protease (if cell wall components were relied upon for an adjuvant effect).

- Although that would be an important step in the further development of this vaccine it would not be necessary for publication of this work if the authors would simply give the readers a better idea of their immunogen by performing SDS-PAGE on their cell wall extract so that a coomassie stain would give a better picture of the protein content and a western blot with their immunized mouse serum would indicate which proteins (if any) actually induced an antibody response. Another possibility that the authors should consider is that the effective immune response was to glycan or peptidoglycan components of the cell wall which is the basis for many traditional bacterial vaccines (vaccine developers these days are focused on protein vaccines because of the ease of searching large genome databases and the ease of making protein immunogens rather than because proteins are the only immunogenic molecules in pathogens).

- This paper would be much richer if some of these issues around what was the immune response that provided the protection observed were discussed.

- On another note, this paper would benefit from better proof reading before submission (or resubmission).

         - There are multiple sentences where a useful “the”, “an”, or verb was omitted.

         - The word “pilot” (as in pilot studies) is repeatedly spelled as “polit.”

         - Sometimes word order is mixed up (“the samples were centrifuged… and poured out supernatant.”) and sometimes words are inappropriately truncated (“cell walls were add” instead of “cell walls were added”).

         - This looks like a lazy use of spell check and grammar check functions without the author actually reading the fixed text to see if it still makes sense.

- All in all, this is an interesting investigation into the development of a much needed vaccine. However, significant changes must be made to make this paper acceptable for publication.

Author Response

# Reviewer 1

Comments and Suggestions for Authors

This article may report an important first step in the making of a needed new vaccine, however, there are significant gaps in the information provided in Materials and Methods and in the analyses performed that make it unacceptable for publication in its current form.

Comment 1: The Materials and Methods section is supposed to give other scientists enough information about the procedures to allow them to repeat the studies in another laboratory. This is done by either giving clear and complete information about (usually novel or uncommon) procedures or references to other publications in which the procedures are clearly described.

Response: Thank you for the comment. Authors’ agree with the reviewer and therefore more information and/or relevant references have been added to materials and methods section accordingly.

Comment 2: In this article the only reference in the entire M&M section is to textbook guidelines which were followed “with appropriate adjustment” undescribed.

Response: The article was already having lot of references and therefore to limit the numbers to fit within the journal requirments, we restricted the references. However, based on reviewer’s comment and to support the methods section, additional relevant references in the M&M section have been added.

Comment 3:  A complicated cell wall protein extraction is described (M&M section 2.3) but while volumes of suspension are given no estimate of the number of cells extracted is given so it is impossible to determine the yield of this procedure.

Response: In this study, an average of 7 g/L of C. parapsilosis cell pellet was obtained which resulted in a total of 660 mg/L of protein extract powder. Later on, the extracted powder was used to evaluate the protein concentration with the help of DC Protein Assay (Bio-Rad) using BSA as a standard. The final yield was around 70 µg per 1 mg/mL of protein extract powder. Additionally, for clear understanding the revised manuscript has been supplement with the amount of cells extracted and also the amount of protein injected in each mice has been cleary mentioned in the method sections {M&M2.3 and 2.9}.

Comment 4: Furthermore, some of the reagents are listed as “protease inhibitor cocktail” or “protein extraction buffer” with no information about their composition. At one point it says that “The (glass) beads were then removed” but it doesn’t say how; that is important information because if they were filtered out then one wants to know the pore size of the filter to have an idea of what else might have been removed or if they were centrifuged out then at what speed so that the reader has some idea of what else might have been removed.

Response: The “protease inhibitor cocktail” was brought from Sigma-Aldrich and this has now been mentioned in the revised protocol. The chemical composition of protein extraction buffer and the method for removing the glass beads has now been detailed in the revised manuscript.

Comment 5:  In the Immunization section (M&M section 2.9) no information is given about the amount of protein in the inoculum which is simply described as “60 ul of extracted cell wall proteins” although later the article states that “a booster dose of the same protein concentration was given” so the authors must know what the protein concentration was. This is important information because most of the 933 C. parapsilosis proteins detected in the cell wall extract by the sophisticated mass spec analysis described were unlikely to have been present in the inoculum at sufficient quantity to induce an immune response.

Response: Authors agree that the sufficient amount of proteins in the extract is the key to induce immune response and therefore, 300 µg of protein extract was injected in each mice. This information has now been detailed to the immunization section.

Comment 6:  Also, it is not stated whether the extract was inoculated with an adjuvant to enhance the immune response to low concentration or poorly immunogenic proteins. If an adjuvant was used, then it is important to name it and detail the amount used. But I say “if an adjuvant was used” because that raises another and perhaps more important issue with this paper.

Response: Authors’ thank reviewer for highlighting this point. We agree that the use of adjuvant in this study would have been key especially for low concentration or poorly immunogenic proteins. However, as repeatedly mentioned this is an intial pilot study to establish the importance of use of the cell wall extracts and therefore no adjuvant was used to enhance the efficacy of the protein extract. In the future study, where specific protein/s will be used, an adjuvants will also be included.

Comment 7: It is well known that pathogen cell walls are replete with molecules that stimulate innate immune responses with adjuvant-like effects (pathogen-associated molecular patterns or PAMPs) and some adjuvants (e.g. MPLA) are actually purified from pathogen extracts, so these authors may have relied on adjuvant-like molecules in their cell wall extract that co-purified with the cell wall protein. In fact, absent information on the proteins present in the extract in sufficient quantity to be immunogenic, it is possible that the only immune response generated was the innate response and there was no appreciable or functional anti-cell-wall-protein antibody response induced. Innate immune mechanisms contribute to the control of fungal pathogens such as C. parapsilosis and the challenge experiment described in this article was performed only two weeks after the booster shot so any innate immunity induced by the inoculations was still active during the challenge and could have been the sole reason for the protection observed. This could be sorted out by repeating the immunization/challenge study with an adjuvant alone control arm (if an adjuvant was used) or a control arm with the cell wall extract treated with protease (if cell wall components were relied upon for an adjuvant effect).

Response: As the study was done without an adjuvant, there is no demand of using an adjuvant alone control arm; whereas we agree that the protease treated cell wall extract would have been ideal control. We will keep this suggestion for future research. For this study, we included the control arm where only saline was injected to avoid any false results. 

Comment 8: Although that would be an important step in the further development of this vaccine it would not be necessary for publication of this work if the authors would simply give the readers a better idea of their immunogen by performing SDS-PAGE on their cell wall extract so that a coomassie stain would give a better picture of the protein content and a western blot with their immunized mouse serum would indicate which proteins (if any) actually induced an antibody response. Another possibility that the authors should consider is that the effective immune response was to glycan or peptidoglycan components of the cell wall which is the basis for many traditional bacterial vaccines (vaccine developers these days are focused on protein vaccines because of the ease of searching large genome databases and the ease of making protein immunogens rather than because proteins are the only immunogenic molecules in pathogens).

Response: Authors’ thank reviewer for the comment and agree that further electrophoresis will validate the results obtained and determine the specific proteins involved in the immunization. As that is the part of the project and will be included in future studies, we compiled the first part of the study in this manuscript. Publication of this part will authenticate and allow us to continue with this study.

Comment 9: This paper would be much richer if some of these issues around what was the immune response that provided the protection observed were discussed.

Response: As suggested the immune response mechanism has been added under the discussion section.

Comment 10: On another note, this paper would benefit from better proof reading before submission (or resubmission).

         - There are multiple sentences where a useful “the”, “an”, or verb was omitted.

         - The word “pilot” (as in pilot studies) is repeatedly spelled as “polit.”

         - Sometimes word order is mixed up (“the samples were centrifuged… and poured out supernatant.”) and sometimes words are inappropriately truncated (“cell walls were add” instead of “cell walls were added”).

         - This looks like a lazy use of spell check and grammar check functions without the author actually reading the fixed text to see if it still makes sense.

Response: The whole manuscript has been edited for grammar, language and typo mistakes.

- All in all, this is an interesting investigation into the development of a much needed vaccine. However, significant changes must be made to make this paper acceptable for publication.

Response: Authors’ thank reviewer for the time and indepth review of our manuscript. All the suggestions have been noticed and addressed while revising the paper.

Reviewer 2 Report

This paper reports on characterized the cell wall immunodominant proteins of pathogenic yeast C. parapsilosis, and evaluated their protection effects in mice. In the general, the authors have carried out an interesting and logical series of experiments, which I found to be quite interesting. However, this manuscript needs professional English editing to be published. There are some details that could use further attention.

1. References are too old, please add articles cited for nearly 5 years.

2. MIC results were further collated to determine drug resistance against drug sensitivity criteria.

3. Fig1 Poor quality, cannot see the abscissa.

4. The CON group in Fig.2 is counted as 0 or as 0 after testing?

5. The statistical results in Fig. 2 are re-described to indicate which two groups are significantly different.

6. Add a ruler to Fig. 3 and describe the pathological changes in detail. It is best to find a professional to compare the same tissue site. If possible, the damaged area can be enlarged.

7. Materials and methods Add strains to subsequent treatment methods ( including biosafety and how to avoid contamination )

8. The description of the protein one by one in the discussion is too empty. Published articles can be combined to find out whether there is an association or function between proteins.

Author Response

# Reviewer 2

Comments and Suggestions for Authors

This paper reports on characterized the cell wall immunodominant proteins of pathogenic yeast C. parapsilosis, and evaluated their protection effects in mice. In the general, the authors have carried out an interesting and logical series of experiments, which I found to be quite interesting. However, this manuscript needs professional English editing to be published. There are some details that could use further attention.

Response: Authors’ thank reviewer for the appreciation and for the comments. As suggested the whole manuscript has been edited for grammar, language and typo mistakes.

  1. References are too old, please add articles cited for nearly 5 years.

Response: Authors’ agree that the references used in the article are old and new references would have been more impactful. However, very little research has been done on C. parapsilosis and even less on cell wall related topic. Most relevant papers were published from a long time ago. This also highlights the demand of the work done in this article to fill the knowledge gap.

  1. MIC results were further collated to determine drug resistance against drug sensitivity criteria.

Response: MIC of the common drugs was determined against all the test strains to screen out the one most resistant strain for further assays in this study.

  1. Fig1 Poor quality, cannot see the abscissa.

Response: Fig 1 has been modified for clear reading.

  1. The CON group in Fig.2 is counted as 0 or as 0 after testing?

Response: The health control group was counted as 0 because there were less than 5 colonies on the plates.

  1. The statistical results in Fig. 2 are re-described to indicate which two groups are significantly different.

Response: Authors thank reviewer for noticing this. The duplicated description has now been removed in the revised manuscript.

  1. Add a ruler to Fig. 3 and describe the pathological changes in detail. It is best to find a professional to compare the same tissue site. If possible, the damaged area can be enlarged.

Response: The scale bar has been added and a more detailed description of the histopathology observation has been added under the discussion section.

  1. Materials and methods Add strains to subsequent treatment methods (including biosafety and how to avoid contamination )

Response: All the strains used in this study were handled following the laboratory SOPs to avoid any contamination and cross-contamination. For biosafety, the protocol was assessed and approved by Institutional Biosafety Committee under the reference number of 20210502Lab. These details have now been mentioned in the revised manuscript. 

  1. The description of the protein one by one in the discussion is too empty. Published articles can be combined to find out whether there is an association or function between proteins.

Response: Thank you for your comment. Although, not much research has been conducted on C. parapsilosis cell wall proteins, but we made effort to compile the data from other Candida species and Saccharomyces cerevisiae to interprate the antigenic potential of various cell wall associated proteins. This study open up a new window for purification of these immunogenic proteins and taking them forward to vaccine development pipeline.

The orthologous of the protein encoded by CPAR2_404800, CPAR2_701270, CPAR2_703940 and CPAR2_805040 are mostly reported for one or the other pathogenic attriburtes in different species Candida and S. cerevisiae and therefore, the presence of these virulence associated proteins in cell wall extract of C. parapsilosis strengthen its immunogenic property. Moreover, we speculate that these proteins are immunogenic in nature and together they triger strong immune response against the C. parapsilosis in the vaccinated group.

All this information has been added to discussion section of the newly submitted manuscript.

Reviewer 3 Report

Authors claimed that they have identified a cocktail of 34 immunodominant proteins from Candida cell wall , using it to immunize immunocompetent Balb/C mice and resulted in lower fungi burden, as shown in Figure 2. They claimed that this cocktail is effective in preventing hematogenous dissemination of candida.

Table 1 and line 210: Authors should briefly review the mechanisms of these fungicides that they used in the materials and methods.  Presumedly these fungicides/fungistatics are mostly targeted on  beta glucan species, although it does not affected the results. In Line 210, you should explain based on what findings that #4112, 4033, 5503 were selected for further study.  But there is no further results of #4033 and 5503 being described.

Table 2, later half (about localization and type) and lines 289-303: regarding CPAR2_404800: in Table 2 you state that it is localized in cell membrane, whereas in line 289-303 (discussion) you state that it is on the cell wall.    These two localizations are quite contradictory to each other, so which one is correct? You are suppose to extract cell wall proteins.

Table 2: Again CPAR2_405130 and 701270 are located in the cytoplasm, and CPAR2_703940 and 805040 are extracellular.  According to your extraction protocol, extracellular proteins should have been removed before extraction, and the cytoplasmic organelle protein are not intended to be extracted.  Have you verified your protocol, or have you seen any literature that succeeded in extracting cell wall proteins using this protocol?

Figure 3: What you show here are not "abscess or microabscess". An abscess is a necrotic focus in which tissue is lysed by proteolytic proteins released from neutrophils. Here I see neither foci nor of few neutrophils.  The resolution of figures is insufficient to claim any PAS stained structures.  The arrows are pointing to intravascular structures most likely blood cells, because they don't look like any of the Candida forms: yeast, pseudohyyphae, or hyphae.  Assuming the histopathology are correct, what you are showing here (48 hours post-vaccination, line 273) and what you are testing the innate immunity to Candida, not the cell wall protein-induced immunity.  This is much faster than the principle of acquired immune responses that we know.

Author Response

# Reviewer 3

Comments and Suggestions for Authors

Authors claimed that they have identified a cocktail of 34 immunodominant proteins from Candida cell wall , using it to immunize immunocompetent Balb/C mice and resulted in lower fungi burden, as shown in Figure 2. They claimed that this cocktail is effective in preventing hematogenous dissemination of candida.

Table 1 and line 210: Authors should briefly review the mechanisms of these fungicides that they used in the materials and methods.  Presumedly these fungicides/fungistatics are mostly targeted on  beta glucan species, although it does not affected the results. In Line 210, you should explain based on what findings that #4112, 4033, 5503 were selected for further study.  But there is no further results of #4033 and 5503 being described.

Response: Authors’ thank reviewer in finding the typo error, which has now been corrected in the revised manuscript. Only strain 4112 was selected out of 12 tested strains.

Table 2, later half (about localization and type) and lines 289-303: regarding CPAR2_404800: in Table 2 you state that it is localized in cell membrane, whereas in line 289-303 (discussion) you state that it is on the cell wall. These two localizations are quite contradictory to each other, so which one is correct? You are suppose to extract cell wall proteins.

Response: Authors are thankful for your comment and highlighting this point in Table 2. In present investigation we used DeepLoc analysis to predict subcellular localization of the identified peoteins. The DeepLoc method doesn’t specifically look at the cell wall. Rather only looks at the likelihood of being in the cell membrane and other compartments (nucleus, cytoplasm, extracellular etc.). So in this case proteins that are likely to be in the cell wall are lumped together as cell membrane predicted (as was done in the case of the reference papers we based the analysis on).

Also, in previous study ALS N domain-containing proteins were found to be localized on the cell surface of C. albicans and Saccharomyces cerevisiae. Therefore, in light of previous publications,  principal of DeepLoc methodology and to avoid any further confusion, we have replaced the localization of ALS proteins (encoded by CPAR2_404800 gene) from cell membrane to cell wall in the newly submitted manuscript.

Table 2: Again CPAR2_405130 and 701270 are located in the cytoplasm, and CPAR2_703940 and 805040 are extracellular.  According to your extraction protocol, extracellular proteins should have been removed before extraction, and the cytoplasmic organelle protein are not intended to be extracted.  Have you verified your protocol, or have you seen any literature that succeeded in extracting cell wall proteins using this protocol?

Response: The protocol we followed was described by El Khoury et al. (2018) [DOI: 10.1371/journal.pone.0194403], and Lee et al. (2014) [DOI: 10.1111/jam.12562] have also reported success in extracting cell wall proteins using a similar protocol. It is also possible that some of the proteins are moonlight proteins that have more than one function and location within the cell. However, these proteins have not been characterized yet in C. parapsilosis. We have added the explanation of moonlight protein to the paper under the discussion section.

Figure 3: What you show here are not "abscess or microabscess". An abscess is a necrotic focus in which tissue is lysed by proteolytic proteins released from neutrophils. Here I see neither foci nor of few neutrophils. The resolution of figures is insufficient to claim any PAS stained structures.  The arrows are pointing to intravascular structures most likely blood cells, because they don't look like any of the Candida forms: yeast, pseudohyyphae, or hyphae.  Assuming the histopathology are correct, what you are showing here (48 hours post-vaccination, line 273) and what you are testing the innate immunity to Candida, not the cell wall protein-induced immunity.  This is much faster than the principle of acquired immune responses that we know.

Response: Authors are thankful for your comment. As suggested, the description for figure 3 has been modified according to the recommendation given by the reviewer.

Round 2

Reviewer 1 Report

The authors have addressed some of the issues I raised in my review of the first submission by supplying required detail to the Materials and Methods section and correctly editing the English usage. However, they continue to assert the computer analysis of their sophisticated mass spec analysis of the protein contents of their cell wall extract has identified the immunodominant proteins present when it has only “predicted” what proteins may be immunogenic. They assert that their immunizations has induced antibodies that have protected against C. parapsilosis infection when they have not demonstrated any antibodies have been induced at all (as they would have if they would perform a simple western blot on the extract with their immunized mouse sera) nor have they considered that the entire protection effect may have been de to the activation of innate immune factors by nonprotein cell wall components in the inoculation. Thus, they continue to make unjustified assertions about what they have demonstrated and they continue to ignore the very real possibility of alternative interpretations of their data. Overclaiming results and refusal to consider alternatives in discussion does not make for an acceptable scientific article.

Author Response

Thank you for giving us the opportunity to submit a revised draft of our manuscript entitled, Candida parapsilosis cell wall proteome characterization and effectiveness against hematogenously disseminated candidiasis in murine model, to Vaccines MDPI. We appreciate the time and effort that you and the reviewers have dedicated to providing your valuable feedback on our manuscript. We are grateful to the reviewers for their insightful comments on our paper. We have been able to incorporate changes to reflect all of the suggestions provided by the reviewers. The changes within the manuscript are highlighted.

Here is a point-by-point response to the reviewers’ comments and concerns.

Reviewer 1:

Comments and Suggestions for Authors

The authors have addressed some of the issues I raised in my review of the first submission by supplying required detail to the Materials and Methods section and correctly editing the English usage. However, they continue to assert the computer analysis of their sophisticated mass spec analysis of the protein contents of their cell wall extract has identified the immunodominant proteins present when it has only “predicted” what proteins may be immunogenic. They assert that their immunizations has induced antibodies that have protected against C. parapsilosis infection when they have not demonstrated any antibodies have been induced at all (as they would have if they would perform a simple western blot on the extract with their immunized mouse sera) nor have they considered that the entire protection effect may have been due to the activation of innate immune factors by nonprotein cell wall components in the inoculation. Thus, they continue to make unjustified assertions about what they have demonstrated and they continue to ignore the very real possibility of alternative interpretations of their data. Overclaiming results and refusal to consider alternatives in discussion does not make for an acceptable scientific article.

Response: Authors agree with the reviewer’s comment that further studies are required to prove some of these claims. However, we repeatedly mentioned that this was a pilot study which advocated the immunogenic potential of C. parapsilosis cell wall proteins against invasive candidiasis in Balb/c mice. Due to the lack of studies in this area and knowledge gap, this paper will be important for future studies to further investigate the isolation and purification of proteins that are participating in generation of immune response against C. parapsilosis infection Balb/c mice. Also, these proteins can be used for development of vaccines and diagnostic markers against Candida infection. Due to lack of funding and resources, it is vital for us to publish this paper to generate more fund and complete the study. However, to avoid some of the over-ambitious claims statements made in the manuscript has been edited to fit within the obtained results.

Reviewer 2 Report

No other comments

Author Response

Thank you for giving us the opportunity to submit a revised draft of our manuscript entitled, Candida parapsilosis cell wall proteome characterization and effectiveness against hematogenously disseminated candidiasis in murine model, to Vaccines MDPI. We appreciate the time and effort that you and the reviewers have dedicated to providing your valuable feedback on our manuscript. We are grateful to the reviewers for their insightful comments on our paper. We have been able to incorporate changes to reflect all of the suggestions provided by the reviewers. The changes within the manuscript are highlighted.

Here is a point-by-point response to the reviewers’ comments and concerns.

 # Reviewer 2

Comments and Suggestions for Authors

No other comments

Response: Authors are thankful for your time and inputs on our manuscript.

Reviewer 3 Report

In section 2.9, you bled the mice on days 0, 7, 14, and 21, but you have not shown any immunity results from these blood samples, neither antibody nor cell mediated immunity.

You have not shown the survival rates of the 3 groups (healthy control, cell wall protein, infection control), therefore the protective effect that you claim is not convincing.

Figure 3 is absolutely not acceptable.  Consult any human pathologist or veterinary pathologist before re-submitted. First, none of the 12 micrographs presented in this figure have clearly shown a yeast form of Candida.  I suggest you use higher magnification to show just one convincing yeast form from each group, and use this figure 3 as a compliment to figure 2.  PAS should stain the yeast form pink-purple. Second, to hastily sacrifice animals at 48 hours post-challenge is not a good way to evaluate the effectiveness.  You should wait, let the disease develop, and see the survival rates as well as histopathology plus fungal burdens, which are true testimony of efficacy.  The literature that you cited may had not been scrutinized by qualified pathologists.

Alternatively, delete the histopathology, and change the manuscript type to "brief communication".

Author Response

Thank you for giving us the opportunity to submit a revised draft of our manuscript entitled, Candida parapsilosis cell wall proteome characterization and effectiveness against hematogenously disseminated candidiasis in murine model, to Vaccines MDPI. We appreciate the time and effort that you and the reviewers have dedicated to providing your valuable feedback on our manuscript. We are grateful to the reviewers for their insightful comments on our paper. We have been able to incorporate changes to reflect all of the suggestions provided by the reviewers. The changes within the manuscript are highlighted.

# Reviewer 3

Comments and Suggestions for Authors

  1. In section 2.9, you bled the mice on days 0, 7, 14, and 21, but you have not shown any immunity results from these blood samples, neither antibody nor cell mediated immunity.

Response: The blood collection on different time intervals was done with an intention to determine the antibody profile; however, due to the limited fund and resources the study was not completed. As mentioned earlier, this is a pilot study and further research based on availability of funds will be done where this blood will be used. In order to avoid the readers’ confusion, methods section has been rewritten to remove this additional information.

  1. You have not shown the survival rates of the 3 groups (healthy control, cell wall protein, infection control), therefore the protective effect that you claim is not convincing.

Response: Thank you for your comment. As suggested, survival rate has been appended in the revised figure 2. 

  1. Figure 3 is absolutely not acceptable. Consult any human pathologist or veterinary pathologist before re-submitted.
  • First, none of the 12 micrographs presented in this figure have clearly shown a yeast form of Candida. I suggest you use higher magnification to show just one convincing yeast form from each group, and use this figure 3 as a compliment to figure 2. PAS should stain the yeast form pink-purple.

Response: We completely agree with the reviewer that PAS staining gives pink-purple colour to yeast cells. As suggested, we consulted a human pathologist and according to her these purplish pink dots are yeast forms and since C. parapsilosis does not form hyphae sometimes we confuse it with blood cells. However, after detailed research of slides and comparative analysis of all the groups of mice we came to a conclusion that these purplish pink structures are Candida cells. We also discussed with the staff members involved in staining procedure and they said that the ratio of the reagents used in the staining has major role in developing deep purple colour and high contrast in the slides. The team also confirmed that these dots are none other than yeast cells and less purple colour is due to the ratio of reagents used while staining procedure.

Round 3

Reviewer 1 Report

The paper has been improved by the use of qualifying language around the assertion that cell wall proteins were actually recognized by antibody responses rather than simply being characterized as immunogenic by a computer program, and some lip service to the need to further characterize the protein “immunogens.” And the fact that this is a “preliminary” study to be used to justify the further funding needed for the necessary further characterization is accepted as an excuse for not performing the suggested SDS-PAGE and western blot analyses for this publication. However, it remains a fact that innate responses triggered by nonprotein components of fungal cell walls (see Thoth et al, “Candida parapsilosis: from Genes to the Bedside” Clin Microbiol Rev. 2019 Apr; 32(2): e00111-18) could explain the “protection” observed without an antibody response against cell wall proteins. The preliminary nature of the work reported does not excuse a refusal to include even the simplest of scientific discussions of alternative explanations of results. Also, the word “conjunctured” in line 390 should probably be replaced by “conjectured.”

Reviewer 3 Report

Authors should consult a anatomic pathologist in his or adjacent institution or hospital on Figure 4.  What authors present in Figure 4 is not in a sufficient magnification to judge where the pointed subject are yeast form of Candida or simply clusters of red blood cells.

In healthy control group, the micrographs did not include "clusters"of  red blood cells to be compared with the other two groups, therefore what arrows  indicate there are most likely red blood cells.

I suggest retake micrographs of cell wall protein group and infection control group at at least 1,000x magnification, and show a clear yeast form of Candida to distinguish it from red blood cells.  

The new figure legend is not what I anticipated.